# Lymphocytes subtyping on H&E slides with automatic labelling through same-tissue stained ImmunoFluorescence images

**Etienne Pochet**                                    ETIENNE.POCHET@SANOFI.COM
*Department of Research and Development*
*Sanofi*
*Gentilly, France*

**Luis Cano Ayestas**                              LUIS.CANOAYESTAS@SANOFI.COM
*Department of Research and Development*
*Sanofi*
*Vitry Sur Seine, France*

**Alhassan Casse**                                ALHASSAN.CASSE2@SANOFI.COM
*Department of Research and Development*
*Sanofi*
*Vitry Sur Seine, France*

**Qi Tang**                                                QI.TANG@SANOFI.COM
*Department of Research and Development*
*Sanofi*
*Bridgewater, USA*

**Roger Trullo** ✉*                                  ROGER.TRULLO@SANOFI.COM
*Department of Research and Development*
*Sanofi*
*Gentilly, France*

**Editor:**

## Abstract

Accurate identification and classification of immune cells within tissue samples are critical for understanding disease mechanisms and predicting treatment responses as a cornerstone for personalized medicine. Traditional histopathology relies on hematoxylin and eosin (H&E) staining, which provides structural context but lacks specificity for immune cell sub-types, preventing pathologists from more precise identification. In contrast, immunofluorescent (IF) staining enables precise targeting of specific markers, but this recently developed technology is very costly and not widely applied in clinical practice yet. In this work, we propose a method to leverage registered pairs of H&E and IF stained images from the same tissue to automatically generate cell type labels for H&E from IF marker expression, allowing for precise identification. In particular, we demonstrate the feasibility of lymphocyte sub-typing from H&E images by training cell-level classifiers to accurately distinguish T-cells subtypes (CD45 / CD3e / CD4 / CD8a). Full code will be made available.

---

*. Corresponding author.

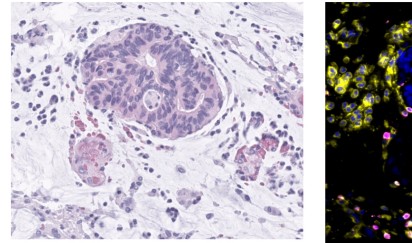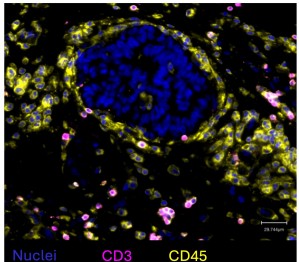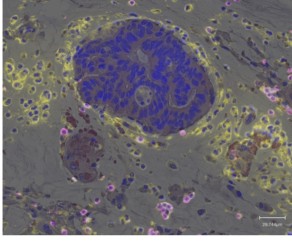

Figure 1: H&E and corresponding IF image in the ORION dataset. Blue, yellow and purple correspond to nuclei, CD45 and CD3 respectively. On the right we display an overlap that shows the cell level alignment between the 2 modalities.

## 1 Introduction

Histopathological analysis relies on H&E stained whole slide images (WSI) as the gold standard for visualizing tissue structures Gurcan et al. (2009). Deep learning research has made great advances in this domain Dimitriou et al. (2019); Komura and Ishikawa (2018); Tizhoosh and Pantanowitz (2018), particularly in cell segmentation Schmidt et al. (2018); Stringer et al. (2021)where recent developments have enabled multiple downstream tasks in digital pathology for diagnosis and prognosis Saltz et al. (2018); Echle et al. (2021); Angell et al. (2020).

While H&E staining is very effective for some tasks, it lacks the discriminatory power needed for precise cell type identification beyond general categories. This limitation makes challenging accurate ground truth labeling and the development of specialized cell classification models. On the other hand, advanced machine learning models may exploit human-imperceptible patterns in H&E images to achieve finer-grained cell type classification.

Accurate identification of precise cell types in H&E images would streamline histopathological workflows, reducing reliance on costly and time-consuming alternatives like Immuno-histochemistry (IHC) and IF Duraiyan et al. (2012). IF imaging offers higher precision and multiplexing capabilities but typically is applied to consecutive cuts which makes cell level alignment with H&E not possible.

In this work, we propose a methodology leveraging a publicly available H&E and IF stained dataset Lin et al. (2023) on same tissue images, enabling semi-automatic generation of cell phenotype labels. Our approach facilitates training classifiers for cell types not extensively studied in literature, such as T-cell lymphocyte sub-types (e.g., CD45-CD3e-CD4-CD8a), critical in immune function and disease. Our work demonstrates feasibility in lymphocyte sub-typing on H&E slides using convolutional neural networks and vision transformers, representing a novel contribution to the field.

## 2 Related Works

In recent years, significant progress has been made in the classification of immune cell types using H&E stained images. Nuclei segmentation in WSI has been extensively studied Hayakawa et al. (2021). Models like HoverNet Graham et al. (2019) and StarDist Schmidt et al. (2018) have achieved high performance, being widely used in numerous applications.

Public datasets like MoNuSAC Kumar et al. (2017) and PanNuke Gamper et al. (2019) provide classification annotations for multiple cell types and nuclei segmentation masks. These datasets allow segmentation models to specialize in cell classification on H&E, with available labels including broad families like epithelial cells, lymphocytes, macrophages, and neutrophils for MoNuSAC, and inflammatory, dead, non-neoplastic, neoplastic, connective for PanNuke. These categories can be reliably distinguished by trained human experts when examining H&E stained samples, as they exhibit distinct morphological characteristics Neumann and Neumann (2021).

Few works have enriched H&E cell classification models with cell-level labels from paired IF images. Reddy et al. Reddy et al. (2022) demonstrated lymphocyte prediction in colorectal cancer (CRC) patients using algorithmically generated annotations from paired H&E and IF images. Similarly, Wee et al. (2023) identified specific sub-types of CD8+ cells on H&E images using labels from co-registered IF images, applied to a private dataset of mouse tissue samples.

Our work innovates by identifying and classifying fine-grained lymphocyte sub-types directly from routine H&E stained images, leveraging a large publicly available dataset. This is achieved through a computational approach that generates granular single-cell training labels using paired multimodal data from H&E and IF imaging. Previous efforts focused on broader immune cell categorizations discernible by human observers based on H&E morphology or suggested lymphocyte subtyping potential using proprietary datasets. In contrast, our methodology studies the ability to automatically estimate lymphocyte subsets, which is not feasible through manual annotation.

## 3 Methods

### 3.1 Dataset

We use the data shared by Lin et al. (2023), who introduced a new approach for high-plex IF and H&E imaging of the same tissue sample. Their platform, ORION, enables the simultaneous capture of 16-18 markers, surpassing the typical clinical research standard of five or six plex. ORION facilitates high-quality IF and H&E imaging, validated through rigorous qualitative and quantitative assessments. The authors shared a dataset of 41 image pairs (H&E and IF) with a resolution of 0.325 microns per pixel from 40 colorectal cancer (CRC) patients. The dataset includes 16 markers relevant to CRC (e.g., CD45, CD3e, CD4, CD8a, CD68, Pan-CK). They demonstrated their approach's potential by using cell-level information from both modalities to model progression-free survival. They also released initial pre-processing results, including image registration between IF and H&E and cell segmentation based on the nuclei marker on IF. An example of the imaging data is shown in Figure 1 and interactive visualizations of all 41 image pairs can be found at `https://www.tissue-atlas.org/atlas-datasets/lin-chen-campton-2023/`.

### 3.2 Semi-automatic dataset generation

#### 3.2.1 Cell labelling

We leverage the IF expressions to generate labels for the H&E images. Visual inspection showed that the registration of both images did not reach pixel-level alignment. Hence, we

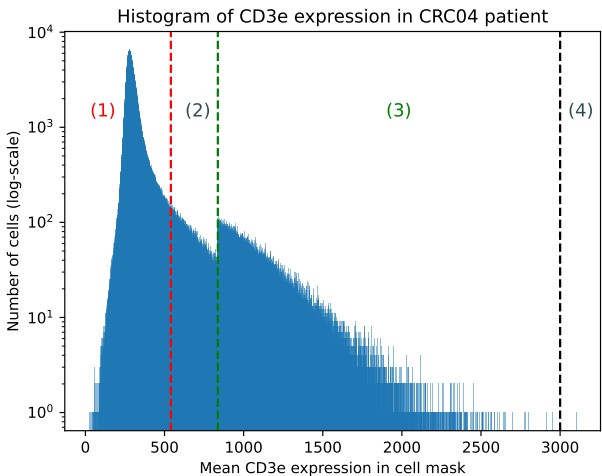

Figure 2: Example histogram of cell expressions and labelling thresholds (Histogram of CD3 marker for patient CRC04). (1) Negative range. (2) Uncertainty range. (3) Positive range. (4) Over-exposure range

decided to focus on generating labels for cell level classification, instead of segmentation mask which is therefore out of the scope of this work.

When analyzing IF images, pathologists visually classify cells based on their marker expressions, effectively applying a thresholding algorithm manually. We hypothesize that this workload can be greatly reduced for the annotator and that many of those cell annotations could be obtained automatically. Instead of selecting a single binary threshold as it is commonly done, we chose to define multiple ranges, to allow for cells to be labelled as positive, negative or unknown, In particular, we define *Negative range*: include expressions low enough that cells can be considered negatives with high confidence. In this range, marker expression is of the same order of magnitude than the background noise; *Positive range*: expression is high enough that cells can be considered positives with high confidence; *Uncertainty range*: Cell expression is in between the background noise and the clearly positive cells. Cells with marker expression in this range are not labelled by the algorithm; *Over exposure range*: cell expression is too high to say with confidence that it is not due to staining artefacts. Cells with expression above this value are not labelled.

Trained pathologists can distinguish cells more accurately by considering local context. However, algorithmically labeling extensive data involves trade-offs. To minimize mislabeling risk, we chose not to label all cells, setting cautious thresholds to enhance dataset signal-to-noise ratio. Whole Slide Images (WSIs) contain millions of cells; labeling even a fraction yields numerous annotations. In this work, an anatomical pathologist established marker thresholds, enabling semi-automatic labeling of image datasets with minimal human intervention. An example of these thresholds is shown in Fig.2

### 3.2.2 Marker selection and labels curation

One of the objectives of our work is to facilitate the analysis of cell populations that contribute to providing data on disease prognosis. Various studies have been published, demonstrating how the population of CD3e+ and CD8a+ cells play an important role in determining the prognosis of various types of tumors Kasurinen et al. (2022); Zeng et al. (2022). Additionally, the creation of immune scores with these subgroups of cell populations is increasingly advocated in different solid tumors as well as in new guidelines for personalized patient treatment Trabelsi et al. (2019); Panahi et al. (2023). Therefore, we decided to focus on T-cells to demonstrate sub-type classification feasibility from H&E images. These sub-types follow a hierarchical structure Lin et al. (2023). In particular, CD4 and CD8a cells are a sub-type of CD3e cells, which are a sub-type of CD45 cells. To further increase the quality of annotations, we curate the labels we generate by filtering out cells that do not match with those rules *i.e.*, allow cells to be labeled as CD3e positive if they are CD45 negative.

We applied the gating and filtering technique to all 41 available image pairs, resulting in the labeling of numerous cells. The analysis revealed a significant imbalance, particularly in the distribution of CD8a and CD4 labels within the dataset, with each representing a very small proportion of positive samples. This disparity is expected due to their characterization as specialized sub-types, which are less prevalent in tissue samples. Specifically, the positive to negative ratios for CD45, CD3e, CD4, and CD8a markers are approximately 0.22, 0.04, 0.006, and 0.006 respectively. These ratios highlight the challenges faced in a task that is already unachievable for trained humans.

## 3.3 Modelling

### 3.3.1 Patch classification

We study this cell sub-typing problem as an image classification task. To classify a given cell, we crop a patch around its centroid coordinates and feed it to an image classifier network. We leverage state-of-the-art (SOTA) pre-trained backbones as feature extractor, and complement them with Multi-Layer Perceptron (MLP) classification head. As we are trying to predict multiple classes that are not mutually exclusive, we can not use a single model trained with multi-class cross-entropy. The simplest formulation of the problem is to train independent binary classifiers for each one of the target types (left Fig.3). These models are trained with binary cross-entropy loss (BCE). We experiment with convolutional neural networks (CNNs) backbones, in ConvNext Liu et al. (2022) and HoverNet. The latter is the standard model in cell segmentation for H&E images. It is composed with a U-Net Ronneberger et al. (2015) feature extractor and pixel-level prediction heads. Therefore, we leverage the features generated by the U-Net and and replace the pixel classification heads with an average pooling followed by the MLP head. We also experiment with Vision Transformers (ViT)Dosovitskiy et al. (2020), using the Phikon weights Filiot et al. (2023). This very recent model has been trained with Self-Supervised Learning (SSL) on thousands of H&E WSIs. It represents a state-of-the-art foundation model for pathology images.

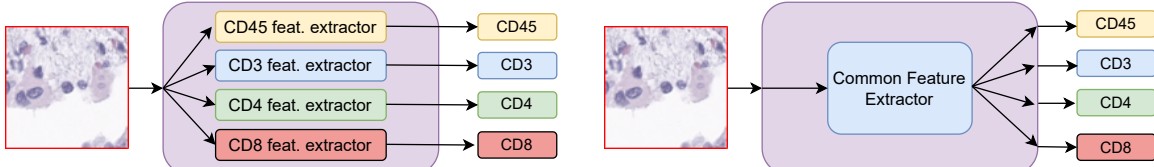

Figure 3: Comparison of different feature extraction strategies: (a) Independent feature extractors are used for each sub-type. (b) In the MTL setup, a common feature extractor is trained for all markers.

### 3.3.2 MULTI-TASK LEARNING (MTL)

The cell sub-types that we selected for this work are closely related, as they are part of the T-cell category. Hence, we hypothesize that joint training on the multiple classes would be beneficial. Therefore, we also experimented with a multi-task learning formulation. In this case, we train models with a shared backbone, in hope that common features can be used to enhance the sub-typing. The common backbone generates the patches features, which are fed to independent classification heads (right Fig.3). Each marker prediction is evaluated with a BCE loss. The total loss is a combination of equally weighted losses of each marker $\mathcal{L} = \frac{1}{4}(\mathcal{L}_{\text{CD45}} + \mathcal{L}_{\text{CD3e}} + \mathcal{L}_{\text{CD4}} + \mathcal{L}_{\text{CD8a}})$

## 4 Experiments

### 4.1 Experimental setup

These experiments are run on a single 16GB V100 GPU. We used AdamW Loshchilov and Hutter (2017) optimizer with a learning rate of 5e-5 and weight decay of 1e-5. The training batch size is 32 with 16 accumulation steps, reaching an effective batch size of 512. We use Timm Wightman (2019) implementation of ConvNeXt Liu et al. (2022) with ImageNet weights and Base size, HuggingFace Wolf et al. (2020) for ViT with Phikon weights Filiot et al. (2023) and TIAToolbox Pocock et al. (2022) implementation of HoverNet Graham et al. (2019) with MonuSACKumar et al. (2017) weights. ConvNext Base and Phikon have 88M parameters while HoverNet has 40M. For ViT, we experiment with resizing the patches to the pre-training patch size of 224 or interpolating position embeddings and report the best results. The classification heads are MLPs with 1 hidden layer and a hidden dimension of 512. In the following results, we fine-tune the entire model, including the pre-trained backbones. We report metrics that account for both the positive and negative class, with Area Under ROC curve (AUROC) and balanced accuracy (BAcc)as recommended in Thölke et al. (2023) for heavily imbalanced classification. The latter is the average of True Negative Rate (TNR) and True Positive Rate (TPR). These metrics allow to evaluate the ability of the model to discriminate between classes while limiting the scaling factor that comes with comparing populations with a 200:1 ratio.

### 4.2 Dataset sampling

To generate train, validation and testing datasets, we randomly split the 41 image pairs into these three categories (25 in training, 7 in validation, 9 in tests). Given the highly

| Model | | | CD45 | | CD3e | | CD4 | | CD8a | |
|---|---|---|---|---|---|---|---|---|---|---|
| Backbone | WBCE | MTL | AUROC | BAcc | AUROC | BAcc | AUROC | BAcc | AUROC | BAcc |
| ConvNeXt (88M) | ✗ | ✗ | **0.96** | **0.89** | 0.90 | 0.81 | 0.90 | 0.68 | 0.88 | 0.54 |
| | ✓ | ✗ | **0.96** | 0.88 | 0.92 | 0.84 | 0.91 | 0.83 | 0.87 | 0.79 |
| | ✗ | ✓ | **0.96** | **0.89** | **0.94** | **0.87** | 0.92 | 0.51 | **0.90** | 0.58 |
| | ✓ | ✓ | 0.95 | 0.85 | 0.93 | 0.85 | 0.89 | 0.80 | 0.87 | 0.76 |
| HoverNet (39M) | ✗ | ✗ | 0.95 | 0.87 | 0.91 | 0.82 | 0.92 | 0.66 | 0.88 | 0.51 |
| | ✓ | ✗ | 0.95 | 0.88 | 0.91 | 0.83 | 0.92 | 0.84 | 0.88 | 0.79 |
| | ✗ | ✓ | **0.96** | **0.89** | **0.94** | 0.85 | **0.93** | 0.50 | **0.90** | 0.53 |
| | ✓ | ✓ | 0.94 | 0.87 | 0.93 | **0.86** | 0.92 | **0.86** | **0.90** | **0.80** |
| ViT Phikon (88M) | ✗ | ✗ | 0.89 | 0.78 | 0.79 | 0.70 | 0.85 | 0.65 | 0.73 | 0.50 |
| | ✓ | ✗ | 0.91 | 0.80 | 0.80 | 0.70 | 0.83 | 0.74 | 0.73 | 0.64 |
| | ✗ | ✓ | 0.91 | 0.82 | 0.82 | 0.73 | 0.82 | 0.50 | 0.76 | 0.52 |
| | ✓ | ✓ | 0.89 | 0.77 | 0.81 | 0.73 | 0.80 | 0.73 | 0.74 | 0.65 |

Table 1: Performance metrics on the testing set on all sub-types of interests.

unbalanced nature of the sub-typing classification task, we decided to resample the training and validation sets to ensure a minimum ratio of positive cells for each markers. In practice, we sampled 20,000 cells for each patient. Half of them are not positive to any markers. In the other half, we make sure that there is at least $\frac{1}{4}$ of positive cells for each of the 4 markers. For the testing patients, we randomly sample 20,000 cells for each patient to match with the real distribution of labels.

### 4.3 Results

#### 4.3.1 OPTIMAL PATCH SIZE

To determine an appropriate patch size, we built a binary classifier of CD45 cells using the ConvNeXt backbone, initial experiments were conducted with various input patch sizes. Results showed that AUROC values improved with increasing patch size until 112 pixels, reaching a peak AUROC of 0.96. Beyond this size, AUROC values remained stable at 0.95 for a patch size of 96 pixels, and gradually decreased to 0.92 for a patch size of 32 pixels. Notably, a patch size of 112 pixels corresponds to approximately 36µm in the images, encompassing a neighborhood of about 3-4 cells around the central cell of interest, which aligns well with the typical size of T-cells (10µm diameter) Lin et al. (2023). This patch size provides a balance between capturing sufficient local context and maintaining focus on the central cell information. All following results are obtained with a patch size of 112 pixels.

#### 4.3.2 MODEL COMPARISONS

We explore various models to subtype T-cells, comparing state-of-the-art backbones: ConvNeXt Li et al. (2022), HoverNet Graham et al. (2019), standard instance segmentation models using the U-Net extractor as described in Sec. 3.3.1, and Phikon, a ViT model pretrained on pathology images. Additionally, we contrast training independent models for each marker with independent backbone tuning against a single multi-task model using a

shared feature extractor and independent classification heads. We also mitigate the impact of high class imbalance by weighting positive samples in the BCE loss, with weights set inversely proportional to the ratio of positive samples in the training set for each marker.

Given the sub-type hierarchy, predicting broader categories like CD45 from H&E is expected to be feasible. CD3e, representing a wider T-cell sub-type, may also be predictable. However, identifying finer sub-types such as CD4 and CD8a is highly challenging due to their scarcity in the dataset. Results in Table 1 confirm this pattern: models achieve better accuracy for CD45 and CD3e compared to CD4 or CD8a. Performance aligns with sub-type hierarchy, with CD45 showing the highest discrimination, followed by CD3e, and the least for CD4 or CD8a.

We observe that CNN-based models consistently outperform ViT models across all markers, indicating convolutional layers better suitability for fine precision tasks and robustness with fewer positive examples. Despite having half the parameters, HoverNet competes well with ConvNeXt, likely benefiting from pre-training on H&E cell segmentation. Moreover, jointly trained models using a shared backbone slightly outperform independently trained counterparts. For instance, with the HoverNet backbone, AUROC increases from 0.91 to 0.94 and BAcc from 0.83 to 0.86 for the CD3e task. This suggests potential in leveraging multi-task learning to enhance training efficiency, despite modest improvements that highlight room for further enhancement in exploiting class similarities.

Balancing category weights in the Binary Cross Entropy loss significantly impacts performance. This adjustment has limited impact on CD45 and CD3 tasks due to their lower dataset imbalance. However, for CD4 and CD8, using weighted BCE dramatically improves balanced accuracy: from 0.5 to 0.86 for CD4 and 0.53 to 0.80 for CD8 with multi-task HoverNet. A balanced accuracy of 0.5 indicates a model assigning the same label to all samples (TPR=0, TNR=1 or vice versa). Weighted BCE effectively enhances learning by emphasizing rare examples. The high imbalance challenges evaluation, where even minor error rates can drastically affect real-world performance. Addressing the trade-off between False Positive and False Negative examples depends heavily on the application. For instance, a tool identifying regions of interest with higher marker expression may prioritize fewer false negatives over more false positives.

Fig. 4 visualizes two regions for all markers using the multi-task HoverNet trained with weighted BCE, offering insights into the model behavior. High accuracy is observed for CD45 prediction, while CD3e predictions show good correlation with marker expression despite numerous false positives. CD4 and CD8 exhibit high recall with visible correlation, yet with considerable false positives due to their rarity. Post-processing, as described in Section 3.2.2, could potentially refine these predictions by enforcing alignment with T-cell sub-type hierarchy.

## 5 Conclusion

This paper presents a methodology for cell classification in H&E images across a broader range of cell types than previously explored in the literature. By leveraging paired H&E and IF images, we semi-automatically generate a large volume of cell-level annotations, ensuring high quality through pathologist input limited to three slide-level thresholds per image pair. Using this algorithmically generated dataset, we train image classifiers to distinguish cells

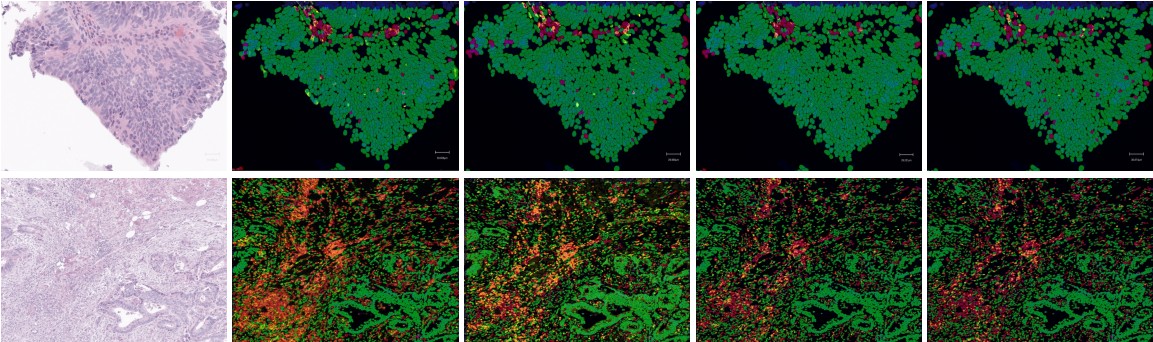

Figure 4: Example of predictions from our model with HoverNet backbone trained through MTL and weighted BCE. From left to right: source H&E region, corresponding IF groundtruth with predictions masks for CD45, CD3, CD4, CD8. Positive predictions in red, negative prediction in green, marker of interest in yellow and nuclei marker in blue. Perfect predictions would display overlapped yellow signal with red masks.

expressing specific markers. We demonstrate promising results in sub-typing T-cells on H&E images, where morphological variations are imperceptible to a trained pathologist. While our methodological approach is simple, it highlights the feasibility of this task.

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
