# OpenReview forum: "Lymphocytes subtyping on H&E slides with automatic labelling through same-tissue stained ImmunoFluorescence images"
_MICCAI.org/2024/Workshop/COMPAYL — COMPAYL 2024_

### Official Review · Reviewer_2o3a · 2024-07-03
**Review for Etienne Pochet**

**Custom Rating:** 4
**Confidence:** 2

**Review:**

Quality:
The quality of the work is overall good.
Pros:
The results obtained from semi-automatically sub-typing T-cells on H&E images are promising.
Significant potential impact on personalized medicine and digital pathology
Cons:
Immediate clinical applicability is not fully demonstrated.
Not very well explained why ViT with higher performance in other domains shows lower performance in this work against the two CNNs.

Clarity:
The clarity of the work is overall okay.
Pros:
The dataset, models, and hyperparameters used are clearly explained.
Cons.
The acronyms are not explained in the text. For example: MTL seems important as it is used in the results table

Originality.
The originality of the work is very good as it presents a novel integration of H&E and IF images using state-of-the-art models.

Significance:
Overall, the significance of this work lies in its potential to transform diagnostic pathology practices, contribute to the advancement of personalized medicine, and improve accessibility to advanced immune profiling techniques.

---

### Official Review · Reviewer_8jD8 · 2024-07-08
**Timely use of IF to predict HE but novelty questionnable**

**Custom Rating:** 3
**Confidence:** 5

**Review:**

This paper investigates the feasibility of predicting lymphocyte populations (CD45, CD3, CD4 and CD8) on H&E images, using multiplexed immunofluorescence images, as a source. 41 images of colorectal cancer from Lin et al are used. Authors employ a multi-task learning approach to perform joint training on multiple lymphocyte classes, compare patch sizes, test CNN and ViT and compare model performance on all cell subtypes. It is also unclear from Figure 1 whether all CD3+ are CD45+ as well (which must be the case). There may be an error in the figure legend with reference to the colors of the cell types. Since CD45 + cells include T-cells and B-cells, the more critical distinction to make is between these 2 classes first, before subtyping T-cells further. It is well-known that lymphocytes can be detected accurately from H&E images, and in colorectal cancers, T-cells are the predominant lymphocyte type found in proximity to the cancer, so the high accuracy for CD3+ cell subtyping explains these observations.  What the cytological aspects are that may help to distinguish CD4 and CD8 from H&E are not hypothesized in this paper.

---

### Official Review · Reviewer_GLuq · 2024-07-15
**Review of Lymphocytes subtyping on H&E slides with automatic labelling through same-tissue stained ImmunoFluorescence images**

**Custom Rating:** 3
**Confidence:** 4

**Review:**

The paper presents evaluation of how well DL can detect different subtypes of T-cells on H&E stained WSIs. The method involves using immunofluorescence (IF) images to produce semi-automatic labels for the four subtypes of T-cells (CD45, CD3e, CD4, and CD8a) which would be challenging to do for a pathologist. However, the step involves a pathologist to manually select threshold on IF images for the 41 WSI and IF pairs used in the study, hence the generalisation of the results is uncertain.  Furthermore, this setup brings a question of how reliable are the semi-automatic annotations since the evaluation is done on the same semi-automatically prepared ground truth labels.

Three algorithms are tested with two different regimes: a binary classification and multi task learning.  The results indicate that transformer performs worse than CNN methods which is not surprising given the low number of image pairs in the study. The contributions of the paper include:
* evaluating several cell classification setups,
* using IF for semi-automatic labelling,
* classifying more subtypes of T-cells  than previously. However the results for additional subtypes are quite poor.

Overall, the paper is well written and easy to follow.

---

### Decision · Program_Chairs · 2024-07-16

Accept